# Interface Optimization of Metal Quantum Dots/Polymer Nanocomposites and their Properties: Studies of Multi-Functional Organic/Inorganic Hybrid

**DOI:** 10.3390/ma16010150

**Published:** 2022-12-23

**Authors:** Xingfa Ma, Caiwei Li, Mingjun Gao, Xintao Zhang, You Wang, Guang Li

**Affiliations:** 1School of Environmental and Material Engineering, Center of Advanced Functional Materials, Yantai University, Yantai 264005, China; 2National Laboratory of Industrial Control Technology, Institute of Cyber-Systems and Control, Zhejiang University, Hangzhou 310027, China

**Keywords:** metal quantum dots, non-conjugated polymer, hybrid organic/inorganic nanostructures, interface optimization, photo-current switching, multi-functional nanocomposite

## Abstract

Nanomaterials filled polymers system is a simple method to produce organic/inorganic hybrid with synergistic or complementary effects. The properties of nanocomposites strongly depend on the dispersion effects of nanomaterials in the polymer and their interfaces. The optimized interface of nanocomposites would decrease the barrier height between filler and polymer for charge transfer. To avoid aggregation of metal nanoparticles and improve interfacial charge transfer, Pt nanodots filled in the non-conjugated polymer was synthesized with an in situ method. The results exhibited that the absorbance of nanocomposite covered from the visible light region to NIR (near infrared). The photo-current responses to typical visible light and 808 nm NIR were studied based on Au gap electrodes on a flexible substrate. The results showed that the size of Pt nanoparticles was about 1–2 nm and had uniformly dispersed in the polymer matrix. The resulting nanocomposite exhibited photo-current switching behavior to weak visible light and NIR. Simultaneously, the nanocomposite also showed electrical switching responses to strain applied to a certain extent. Well-dispersion of Pt nanodots in the polymer is attributable to the in situ synthesis of metal nanodots, and photo-current switching behavior is due to interface optimization to decrease barrier height between metal filler and polymer. It provided a simple way to obtain organic/inorganic hybrid with external stimuli responses and multi-functionalities.

## 1. Introduction

Currently, multi-functional nanocomposites and flexible devices have received more and more attention and obtained excellent signs of progress in many aspects. However, integrating their Ffmulti-functionalities or intelligent properties, and exploring more wide applications are still significant challenges. The responsive materials can respond to external stimuli, such as light, electric, magnetic fields, temperature, pH, heat, and pressure. These materials can convert the different energy forms to measurable physical signals or mechanical energy. Reviewed current relative references, the development of flexible devices mainly depends on polymer-relative functional composites. These polymers generally contain some σ-σ bonds due to the easy self-rotating of polymers holding σ-σ chemical bonds. Polymers have played a crucial part in designing and applying flexible devices and multi-functional nanocomposites.

Compared with inorganic functional nanocomposites, functional polymers-based nanocomposites are relatively minority and mechanism complex from the electronic level. Organic functional materials involve two classes: conjugated small molecular or polymers (electron delocalization); the other is functional materials filled polymer matrix (composite). Generally, conjugated materials have good conductivity (π-π electron conjugated delocalization) and poor flexibility due to lacking self-rotation of σ-σ chemical bonds. For non-conjugated polymers, even if they have good flexibility, their carrier-sharing property is very poor due to the existence of a large number of localized states. Functional nanocomposites filled in polymers composites are good candidates for developing flexible devices because of the simple process technology and diversification of materials design. Improving optical, electrical, or photoelectrical properties and simultaneously enhancing the flexibility of materials is a contradiction from the designing viewpoint of composite materials. It needs interface optimization to decrease barrier height between fillers and polymers for carriers transfer and compromise between mechanical and functional properties. Its interface physics between polymers and functional nanomaterials is also fairly complex. Good dispersion of nanocomposites in the polymer matrix is the prerequisite to obtaining excellent measurable physical signals for polymer-based nanocomposites since the flexible polymer is an insulating material. Otherwise, some electroactive polymers (EAPs), including electrostrictive materials, conducting polymers, ionic polymers, piezoelectric polymers, and dielectric elastomers, are also suitable for the construction of flexible, intelligent devices. Therefore, PVDF poly-(vinylidene difluoride) is one of the good fundamental candidates for developing multi-functional materials and different flexible devices. PVDF is an interesting polymer containing fluorine because of its outstanding properties, such as piezo-, pyro-, ferroelectric properties, thermal stability, its ability to bend and stretch, and chemical inertness to solvents and acids. Although PVDF-based polymers composites are considered attractive for applications in flexible memories, electronic skins, energy transducers, energy-harvesting devices, nonvolatile memories, and multi-functional portable sensors, other functional properties of PVDF still need to be explored by the micro-structure tailor of nanocomposites for applications in multi-disciplinary fields.

To functionalize PVDF or widen its applications, PVDF/polymers blends, or nanomaterials-filled PVDF-based nanocomposites are still the first choices. For illustration, some typical nanocomposites and examples are introduced for multi-disciplinary fields, such as PVDF/Bi_2_Al_4_O_9_/RGO [1], PVDF/AlO/R-GO [2], kaolin/PVDF-HFP [3], PVDF-HFP/SiO_2_ [4], P3HT/PVDF-HFP [5], PVDF/CsPbBr_3_ perovskite [6], titania nanosheets/PVDF [7] nanocomposite, PVDF/polymer blending composite [8], peptide/PVDF composite [9], polyvinylidene fluoride (PVDF) nanofiber with controlled morphology [10], electrospinning PVDF/cellulose nanofibers [11], etc. These nanocomposites are widely used in energy harvesting, energy storage [12,13], nanogenerators, dye-sensitized solar cells, capacitors, and lithium-ion batteries. Chen and co-workers [14] introduced the applications of PVDF-based ferroelectric polymers in flexible electronics. Gao and co-workers [15] constructed flexible displays with ultrathin multi-functional graphene/PVDF layers. Xiao and co-workers [16] designed a fish-like robot with graphene/PVDF bimorph actuation materials. Yu and co-workers [17] constructed nanofibrous PANI/PVDF strain sensors with highly stretchable and conductivity by electrospinning.

To enhance the dielectric, ferroelectric, and piezoelectric properties, some nanomaterials/PVDF blends are also an effective approach. Some typical nanocomposites are as follows: Fe-doped ZnO/PVDF-TrFE composite films [18], GO/PVDF/ZnO composite membranes [19], PVDF/PVDF-TrFE blended films [20], PVDF-based copolymer and terpolymer [21], PVDF/ZnSe and PVDF/Cu-ZnSe [22], MWCNTs/PVDF [23], electrospun PVDF/cellulose nanofibers [24], Au NPs/PVDF nanocomposite for nerve tissue engineering applications [25], PVDF/BaTiO_3_/carbon nanotubes nanocomposites [26], ZnMn_2_O_4_ particles filled PVDF-based composites [27], and PVDF/ZnO nanofibrous composites [28]. Zhao and co-workers [29] introduced the high lithium salt content of h-BN nanosheets/PVDF-based solid-state composite polymer electrolytes. Panda and co-workers [30] reported the filler dependent on microstructural and optical properties of PVDF/GO nanocomposites. Ramanujam and co-workers [31] explored the effect of casting solvent on the structure, electrical, and thermal behavior of polyvinylidene fluoride (PVDF)/carbon nanofiber (CNF) hybrid nanocomposites. Verma and co-workers [32] reported the photo-piezocatalysis properties of electrospun PVDF/WS_2_ membrane. Su and co-workers [33] obtained the high-performance piezoelectric composites via β phase programming. Roth and co-workers [34] prepared the ferroelectric PVDF-TrFE films using a nanoscale polarization controlling approach. Pusty and co-workers [35] utilized the oxygen-vacancy-induced self-poling, enhancing the piezoelectric property of the W_18_O_49_/PVDF nanocomposite. Sharma and co-workers [36] reviewed the progress on PVDF-based nanocomposites, focused on the applications in energy harvesting and sensing. Saxena and co-workers [37] introduced a comprehensive review of fundamental properties and applications of poly(vinylidene fuoride) (PVDF), and so on.

In the application of the environmental field, PVDF-based nanocomposites are also one of the most extensively developed. Some typical applications include the removal of heavy metals [38,39], photocatalytic applications [40,41,42,43,44], nano-filtration [45,46,47,48,49,50,51,52,53], carbon dioxide absorption [54,55], oil/water separation [56,57,58,59], etc.

From here, we can see that the nanomaterials/PVDF blend is a simple and effective way to modify PVDF materials and widen interdisciplinary applications. To improve their physical and chemical properties, the interface and the relatively fundamental physical issues of nanomaterials/PVDF are still worth exploring due to the properties of nanocomposites enormously depending on the interface interaction between fillers and polymer matrix. Especially in the photo-functional materials fields, there are still some critical issues of material physics, photophysics, photocarrier dynamic, and complex interfaces needing further exploring or optimization to avoid the recombination of photo-generated electrons/holes.

Combined with the above-mentioned progress, the properties of PVDF-based nanocomposites are mainly put on piezoelectric, dielectric, and pyroelectric properties. Regarding the photoelectrical properties of PVDF-based composites, there is a slight reference. To improve the behaviors of electrons or holes of PVDF, nanofillers and polymer blends and their interaction can intensify the in-built fields of nanocomposites for charge separation and transfer. Simultaneously, the strong interaction between nanomaterials and polymers can tailor the chemical properties or physical properties of PVDF-based nanocomposites and widen more potential applications in interdisciplinary fields. Since the contribution of free electrons or holes in materials on the chemical and physical properties is significant.

Compared with other kinds of functional nanomaterials, metal nanostructures have rich electrons and surface plasmon properties. They have had extensive applications in many devices for controlling the concentration and behaviors of carriers. Nanostructured noble metals, such as Ag, Pt, Au, Pd, etc., have good optical properties and are applied in the construction of several sensors based on their surface plasmon resonance depending on their size, dimensional, and distance [60,61,62,63,64,65]. In the field of new energy, especially fuel conversion, water-splitting H_2_ photoproduction, and biofuel cells [66,67,68,69,70] have also been widely used. These noble metals are also utilized as typical catalysts [71,72,73,74], participating in some essential chemical reactions. In the applications of drug release controlled by light [75], NIR emissive metal clusters [76], optoelectronic diagnostics as electrochemically active biofilms [77], sensing fields [78], metal nanostructures with different sizes and dimensional showed strong surface plasmon resonance and surface-enhanced Raman scattering. Low-dimensional noble metals with high aspect ratios are needed to enhance the surface plasmon resonance properties in NIR because the degree of red-shift to NIR strongly depends on the high aspect ratio. Compared with low-dimensional metals, the synthesis of metal nanoparticles is simple and low-cost. It still needs surfactants or organic capping agents to prevent aggregation of metal nanoparticles and to improve the stability of nanoparticles. To overcome this bottleneck, Pt precursor (chloroplatinic acid) was reduced with reducing agents, the reducing agent was placed in an oil-soluble polymer matrix, and the phase inversion approach was applied in the synthesis of nanocomposite to obtain an excellent dispersion effect in this study. Reaction sites were located at the interface between the oil phase of the PVDF solution and the water phase of a drop of chloroplatinic acid solution. The interface between metal and polymer is crucial for carriers’ photo-generated injection and transfer, decreasing the recombination of electrons and holes. Metal nanostructure-filled polymers are also typical kinds of organic/inorganic nanocomposites. Interface optimization of metal/semiconductor/polymer is a unique key factor in enhancing the performance of different devices. This study is focused on the interface of metal nanostructures/organic semiconductor/non-conjugated polymer and its carriers transfer by light inducement since the interface between metal and the non-conjugated polymer is more complex.

For many years, Ma and co-authors [79,80,81,82,83] have focused on the microstructures of organic/inorganic multi-functional nanocomposites and their functional properties tailoring. The materials system involved in conjugated polymers, metal oxides, metal sulfides, non-conjugated polymers, conjugated small organic molecules, carbon nanomaterials (CNTs, graphene oxide, carbon nanodots, other carbon nanostructures), nanodots, heterostructures, and organic/inorganic hybrid nanostructures. The photoconductivity of nanomaterials was reported by Ma and co-authors [84]. In the most recent 5–10 years, the interests of Ma transferred to photoelectric properties studies to weak lights of different wavelengths for exploring the relationship between micro-structures of nanocomposites and their properties. Since many nanocomposites have multi-functionalities based on the interaction between light and matter, the enhancement of photoelectric properties can be tailored by band-gap engineering, doping, defects engineering, and interface optimization of nanocomposites. Compared to a conjugated polymer system, a more complex mechanism of nanomaterials/non-conjugated polymer hybrid is present for the photogenerated carriers transfer due to the presence of a chemical state and localized state. More energy from the carriers of photogeneration would be dissipated by the scattering of electron–electron and electron–phonon, and the recombination of electron and hole is also severe. Therefore, interface optimization of nanomaterials/non-conjugated polymer hybrid is crucial to suppress the recombination of electrons/holes, and promote the production, separation, and transfer of photo-generated carriers. At the 2017 Chinese Materials Conference [85], Ma and co-authors tried the dispersion of GO (graphene oxide) solution in oil-soluble polymers with an oil/water interface diffusion. Sound dispersion effects were obtained. However, the photo-current response property of nanocomposites is minimal due to the presents of a large number of traps, chemical states, localized states, and interface states near the interfaces of nanocomposites. Photo-generated electrons were easily trapped by the defects mentioned earlier for a non-conjugated polymer system. In a previous report [86], our focus was put on the addition of small organic molecules to enhance the stability of nanoparticles, avoid aggregation, and their photo-current response properties characterization of nanocomposites. In this paper, the emphasis was placed on nanomaterials filled non-conjugated polymers system and light–matter interaction. Well-dispersion of Pt nanodots/polymer nanocomposite was obtained similarly. Two kinds of interface optimization were considered: one kind of interface between metal nanostructure and conjugated polymer for “hot carriers” injected in conjugated polymer, and the other interface between the conjugated polymer and non-conjugated polymer for photo-generated carriers transfer in the polymer. Photo-conductive responses to weak visible light, 808 nm selected typical NIR, and tentacle sensitivity were examined for potential applications of some flexible devices controlled by light and force. Some good results were obtained, and their mechanisms of photo-current generation, transfer, and tentacle sensitivity are discussed based nanomaterial-filled polymers system.

## 2. Experimental Details

*Materials.* HPtCl_4_ (AR), aniline (AR), and DMF (Dimethyl Formamide) were purchased from Sinopharm Chemical Reagent Co., Ltd., etc. Shanghai City, China. PVDF (FR902) was purchased from Shanghai 3F New Material Co., Ltd., Shanghai, China.

Preparation of PVDF solution. 10 g PVDF, and 200 mL DMF were added to a 500 mL glass vessel. PVDF solution (concentration of PVDF is about 0.05 g/mL) was obtained, then 1 mL aniline was added to PVDF solution (concentration of aniline is about 0.005 mL/mL).

Preparation of HPtCl_4_ solution. 1 g HPtCl_4_, 1000 mL H_2_O was added in 1000 mL glass vessel, then HPtCl_4_ solution (concentration of HPtCl_4_is about 1 mg/mL) was obtained.

Synthesis of Pt/PVDF nanocomposites at the interface of oil/water. In the experiment, about 50, 100, and 150 mL HPtCl_4_ solution were added step by step drop-wise to 30 mL PVDF solution (the concentration of PVDF is about 0.05 g/mL) containing aniline (the concentration of aniline is about 0.005 mL/mL), respectively. Precipitation of PVDF/Pt appeared slowly. The resulting nanocomposite was filtrated and dried at room temperature, then 50-100 mL DMF was added, and the Pt/PVDF solution was formed for later use.

Morphology observation with SEM. The SEM observation was carried out with ZEISS-300 (field emission scanning electron microscopy). The Pt/PVDF solution was cast on an aluminum foil substrate, dried at room temperature, and then sputtered with a thin layer of Au on the surface for the SEM observation.

Morphology observation with TEM. The TEM observation was carried out with JEM-1011. The Pt/PVDF solution was cast on copper mesh coated with carbon film, and dried at room temperature for 30 min.

Measurement of UV-Vis-NIR Spectrum. The UV-Vis-NIR spectroscopic measurements were made with the help of a TU-1810 spectrophotometer in the wavelength range of 200–1100 nm, and the samples used Pt/PVDF solution.

Measurement of FTIR Spectra. FTIR (Fourier transform infrared spectroscopy) spectra were taken with KBr and recorded on IR Prestige-21 Fourier transform infrared spectrometer. Sample and KBr powders were mixed and pressed into a small slice, then dried at room temperature for measurement. Before the measurement of FTIR for the sample, the background baseline was scanned to remove some interference in the air.

Photo-responses of nanocomposite to visible light and NIR. The Pt/PVDF solution was cast on the inter-digital electrodes of Au on a flexible PET (polyethylene terephthalate) substrate. After drying, the photo-conductive responses of Pt/PVDF solid film (without any electrolyte) to weak visible light (20–25 W) and 808 nm NIR with low power were carried out to record electrical conductivity changes with or without illumination using LK2000A Electrochemical Work Station from LANLIKE Chemistry and Electron High Technology Co., Ltd. (Tianjin, China) applied 1 V DC bias. The structure of the electrodes is similar to that of the reference [86].

Tentacle sensitivity examination of nanocomposite to force. The Pt/PVDF solution was cast on the A4 paper (about 0.5 cm × 1 cm). After drying, several Ag fibers were adhesion (using conductive adhesive) to the film as electrodes, and electrical responses were preliminarily recorded with LK2000A Electrochemical Work Station from LANLIKE Chemistry, and Electron High Technology Co., Ltd. (China) under different compression forces (such as 1, 2, 5, 10, 20, 50, 100 g weight of balance) applied 1 V DC bias.

## 3. Results and Discussion

It is well known that PVDF is one type of polar polymer due to the presence of F atoms (strong electron acceptor), which could be dissolved in a strong polar solvent, such as DMF. DMF also would be mixed with water. Water is a poor solvent for PVDF. Therefore, when a small amount of water comes into contact with PVDF solution (DMF), precipitation of PVDF at the water interface and DMF occurs since PVDF does not dissolve in water.

When a drop of HPtCl_4_ solution contacts with PVDF solution (DMF) containing aniline, chloroplatinic acid is reduced into Pt, and aniline is oxidized into polyaniline (PANI), precipitation of PVDF at the interface of water and DMF occurs. Pt nanoparticles were coated with a small amount of PANI, uniformly dispersed in the PVDF matrix. The presence of PANI not only prevents the aggregation of Pt nanoparticles as a surface modification agent and stabilizing agent but also enhances the dispersion effects of Pt in the PVDF matrix due to good compatibility and the presence of hydrogen bonding between PANI and PVDF. This idea was inspired by Sajid Fazal and co-workers [87] introducing gold nanoparticles synthesized using a cocoa extract as a reducing and stabilizing agent, avoiding the aggregation of Au nanoparticles. In this study, Pt nanoparticles were synthesized using aniline as a reducing agent, and the resulting polyaniline acted as a stabilizing agent coating the surface of Pt nanoparticles. The comparative optical photos of PVDF and PVDF/Pt nanocomposites synthesized are shown in Figure 1.

Figure 1 shows that the appearance of PVDF and precipitation of PVDF/Pt nanocomposite at the interface of water and DMF are different. The color of precipitation of PVDF at the interface of water and DMF results from the formation of PANI, which belongs to the PVDF/PANI/Pt nanocomposite. PANI acted as a stabilizer and interfacial compatibilizer between the Pt nanoparticle and PVDF matrix. The PVDF/PANI/Pt nanocomposite holds two kinds of interface tailoring, i.e., PANI/Pt interface and PVDF/PANI interface. Under light irradiation, Pt nanoparticles produce “hot electrons,” and some of those electrons have sufficient energy to overcome the Schottky barrier, and be injected into the LUMO level of PANI. Since PANI is a conjugated polymer, the carriers were shared in the PANI layer. However, PVDF belongs to a non-conjugated polymer, a type of insulating material. The transfer of these carriers was blocked. Therefore, the interface barrier height of the PVDF/PANI interface was optimized by adjusting the amount of PVDF added for charge tunneling or hopping. The presence of the F atom as a strong electron acceptor can also promote free electrons in the polyaniline layer to tunnel the interface of PANI/PVDF. According to the conductive mechanism of nanomaterials/polymer nanocomposite [88,89,90,91,92], the Pt/PVDF nanocomposite comprises an intricate network of conducting and insulating components, and its electrical conduction in composite systems is determined by two mechanisms, i.e., percolation in a continuous conducting channel and tunneling between isolated conducting Pt nanodots. The thickness of the PVDF layer of nanocomposite is the critical parameter to carrier transfer [88,89,90,91,92,93]. Otherwise, the electron would be trapped by the localized state of the PVDF layer. It is not easy to transfer in the non-conjugated polymer. The tunneling current (including direct tunneling and hopping) is a function of the thickness of PVDF. It is vital depending on the distance of nanoparticles and uniformity between PANI and PVDF.

The representative SEM images of Pt/PANI/PVDF nanocomposite are shown in Figure 2.

Figure 2 shows that a nano/micro-combined hierarchical structure is observed for Pt/PANI/PVDF nanocomposite. From the low magnification of SEM (Figure 2b), the whole surface includes many nanoparticles. The size is about one μm or so, near that of a drop size of HPtCl_4_ solution, due to the reaction site located surrounding a drop of HPtCl_4_ solution. However, these μm size particles are composed of a significant number of small nanoparticles from the high magnification of SEM (Figure 2a). Because the surface of nanoparticles coats a layer of polymer, it needs a TEM image to confirm further. The representative TEM images of Pt/PANI/PVDF nanocomposite are shown in Figure 3.

Figure 3 shows that many Pt nanoparticles were dispersed in the polymer matrix. The size of Pt is about 1–2 nm, and they are uniformly distributed in the polymer matrix. Excellent dispersion effects were obtained, and aggregation of Pt nanoparticles was effectively avoided.

The FTIR of Pt/PVDF nanocomposite is shown in Figure 4.

As shown in Figure 4, the bands at 839, 1281, and 1403 cm^−1^ corresponded to the polar β phase of PVDF. The amorphous phase is characterized by two bands at 475 cm^−1^ and 872 cm^−1^. The 762 cm^−1^ is attributed to the presence of the α phase of PVDF.

The UV-Vis-NIR of Pt/PVDF nanocomposite was examined. The results are shown in Figure 5.

Figure 5 shows that the absorbance of Pt/PVDF nanocomposite covered the whole region of visible light and extended to NIR (near-infrared). The edge of the band exceeded over 1000 nm. The absorbance of 565 nm or so is the result of Pt nanoparticles surface plasmon resonance. Some absorbance in NIR is mainly caused by the presence of PANI in the PVDF matrix, including absorbance of polarons, bipolarons, π-π* electronic transition, polaron-π electron transition, etc., in conducting polymers. The red shift was induced by scattering of the interface between Pt dot and PANi and PANI/PVDF interface. The effect of the content of HPtCl_4_ adding (wt. 3.22, 6.25, 9.09%, respectively) on the UV-Vis-NIR curve is minimal. Therefore, it is expected that the obtained Pt/PVDF nanocomposite would be more efficiently utilized by the visible light and NIR. The good absorbance of Pt/PVDF nanocomposite in the broadband light spectrum range results from light/nanocomposite interaction. Subsequently, the photo-dynamic process, such as free electrons/holes generation, recombination, etc., would strongly depend on its microstructure and interface contact of Pt/PVDF nanocomposite. It needs to control the microstructure and interface of Pt/PVDF nanocomposite.

Figure 5 shows that the Pt/PVDF nanocomposite had good absorbance in visible light and NIR. Therefore, the photo-conductive properties of nanocomposite to weak visible light and 808 nm NIR were examined in the experiments. Although PVDF holds outstanding piezoelectric, pyroelectric, and dielectric performance, it is almost a little photo-electrical signal due to lacking free electrons and holes under the radiation of light. For Pt/PVDF nanocomposite, it is expected to obtain a photoelectric response because Pt nanoparticles have strong surface plasmon resonance in the visible light region. In this study, the structure of the photo-detector prototype device consisting of an organic/inorganic nanocomposite on PET film substrate and Au gap as electrodes, similar to that of reference [94], which reported that quartz acting substrate and Au gap as electrodes. The results indicated that the photoelectric response signal depends strongly on the contents of Pt nanoparticles. It was found that when the content of HPtCl_4_ adding was 3.22, 6.25% wt., it was challenging to obtain a photoelectric response signal under visible light, and the repeatability was also poor. This illustrated that low-powder light could excite the nanocomposites to produce free electrons and holes. However, the movement of these free carriers was blocked in organic/inorganic nanocomposites with high content of PVDF. As the PVDF content increases, the nanocomposite shows little photo-current switching behavior. The PVDF, as an insulating material, dominates the electrical conductivity due to its high contents. The organic/inorganic hybrid no longer exhibits photo-current switching behavior. When the content of HPtCl_4_ adding was increased to 9.09% wt., the photo-current of Pt/PVDF solid film (without any electrolyte) was increased dramatically on exposure to weak visible light, and the ratio was about 5–6. When the visible light is turned off, the photo-current goes back to the original baseline, and its reproducibility is also excellent. The response time is about 10 s, and the recovery time is about 30 s. It illustrated that the carriers photogenerated could travel freely in the PVDF layer. When the Pt/PVDF nanocomposites were excited with weak 808 nm NIR of 200 mW, Pt/PVDF nanocomposites containing different contents of Pt exhibited similar photo-current responses. When it is further decreased excitation power of incident light, such as using light of 100 mW 808 nm NIR, Pt/PVDF nanocomposites with low contents of Pt (HPtCl_4_ adding was 3.22, 6.25% wt.) showed poor photoelectric responses, and poor repeatability due to the poor uniformity of nanocomposite with high content of PVDF. Pt/PVDF nanocomposites with contents of Pt (HPtCl_4_ adding was 9.09% wt.) still exhibited good photo-responses to 50, 10 mW 808 nm NIR. The response time is about 18 s, and the recovery time is about 6 s. The ratio of On/Off is two orders of magnitude for 200 mW 808 nm NIR. Even if the powder of excitation light is lowered to 10 mW for 808 NIR, the ratio of On/Off is still about two or so. Therefore, with increasing the content of PVDF added in the nanocomposite, the distance between Pt particles would be more prominent, and the photo-generated carriers sharing is blocked. Based on the analysis of the above experimental results, we found that the photo-conductivity of the nanocomposite to visible light is the result of Pt nanoparticles surface plasmon resonance. Under visible light irradiation, Pt nanodots produced “hot electrons”. Some of those electrons have sufficient energy to overcome the Schottky barrier (interface between Pt and PANI) and be injected into the LUMO level of PANI. Since PANI is a conjugated polymer, these carriers were moved freely in the PANI layer. However, PVDF belongs to a non-conjugated polymer. As an insulating material, the carrier sharing was performed according to the mechanism of charge tunneling or hopping, which depends on the thickness of the PVDF layer or the distance between Pt nanodots. In the NIR region, the contribution of photo-current of nanocomposite is the main result of HOMU and LUMO level transformation of PANI. Under NIR light irradiation, excited electrons from the HOMU level jumped into the LUMO level of PANI. Then, these free carriers were transferred in PVDF according to charge tunneling or hopping way. Otherwise, the strong interaction between Pt and PANI promoted the separation of photo-generated electrons/holes due to the narrowing of the band-gap width of PANI. Therefore, the Pt/PANI/PVDF organic/inorganic hybrid showed good photo-current responses to NIR.

The representative results are shown in Figure 6, Figure 7 and Figure 8.

Figure 6, Figure 7 and Figure 8 showed that Pt/PVDF nanocomposite was more sensitive to 808 nm NIR than that weak visible light. It is well-known that 565 nm or so is the result of the Pt nanoparticle’s surface plasmon resonance. The prepared nanocomposite holds a small amount of PANI; PANI generally has good absorbance in NIR due to the presence of polarons and bipolarons in a conjugated polymer. However, producing free electrons and holes for PANI alone is still tricky with the low power of excitation light of 808 NIR. The Pt/PVDF nanocomposite exhibited excellent photo-conductive sensitivity to 808 nm NIR. Maybe the results of solid interaction between Pt and PANI promote the separation of electrons and holes and narrow the band-gap width of PANI. Further, 808 nm NIR has enough energy (hv) excited electrons from the HOMU level to jump into the LUMO level of PANI. These free carriers can overcome the interface barrier height of PANI/PVDF to tunnel or hop.

The repeatability of photo-response to 808 nm NIR was also examined. Good results were obtained. The representative results are shown in Figure 9.

As shown in Figure 9, the Pt/PVDF nanocomposites exhibited excellent photoelectric responses and good repeatability to weak 808 nm NIR after optimizing the content of Pt. These responses are mainly attributed to the superb dispersion effects of Pt nanoparticles in the PVDF matrix and interface contact. It illustrated that the photo-induced electrons and holes in the Pt/PVDF nanocomposite were produced very quickly to weak visible light and NIR. The resulting nanocomposite prolonged the lifetime of photo-induced charges and avoided the recombination of electrons and holes. The photoelectric signal was measured easily. It is expected that the Pt/PVDF nanocomposite will show suitable activities with weak visible light and NIR (near-infrared). On the basis of fundamental piezoelectric, dielectric, and pyroelectric properties, the photoelectrical properties of PVDF-based composites were enhanced clearly. The Pt/PVDF nanocomposite exhibited multi-functionalities. The composite could have potential applications in a flexible light detector to NIR, an intelligent composite film with external stimuli responses, electronic skin, light sensors [94] or actuators, information storage [95,96,97], membrane separation with photocatalysis properties, and multi-functional nanocomposite, etc.

Based on the sea/island model of a nanomaterial-filled polymer system, the nanomaterials acted as islands, and the polymer matrix looked like a sea. Polymers have viscoelastic properties similar to that of unideal springs. Pt nanodots were interconnected with polymers. When an external force was applied to the nanocomposite, the conductivity of the nanocomposite would change naturally due to the stretching and contraction of the polymer. In this study, Pt/PVDF nanocomposite belongs to the organic/inorganic hybrid. PVDF, as its non-conjugated polymer component, has good flexibility. When an external force was applied to organic/inorganic nanocomposite, the deformation of the polymer occurred. The distance between Pt particles would be longer, and the conductivity of the nanocomposite was decreased. If an external force were removed from organic/inorganic hybrid composite, its deformation would also recover. The distance between Pt particles would be shorter, and the conductivity of the nanocomposite would be increased. Therefore, the tentacle sensitivity examination of nanocomposite to compression force of about 50 g was carried out. Preliminary response results are shown in Figure 10.

As shown in Figure 10, when an external force of about 50 g was applied to organic/inorganic hybrid composite, the distance between Pt particles would elongate, and the conductivity of the nanocomposite decreased quickly. When an external force of about 50 g was removed from the organic/inorganic hybrid composite, the distance between Pt particles would shorten, and the conductivity of the nanocomposite increased slowly. The response time is about 12 s, and the recovery time is 438 s. The slow recovery rate is the result of the viscoelastic property of the polymer. The chain segment of polymer recovery needs a long time, depending on the molecular structure and cross-linking degree of the polymer. Appropriate chemical cross-linking of polymer can enhance the response rate and recovery rate. The best candidate is the chemical cross-linking of rubber-like materials.

The effects of different external forces on the tentacle sensitivity of Pt/PVDF nanocomposite were also examined. The results are shown in Figure 11.

Figure 11 shows that the tentacle sensitivity of Pt/PVDF nanocomposite depends on the external force applied. It would be potential applications in tentacle sensors [98,99]. Therefore, this method is a simple way to obtain organic/inorganic hybrid materials holding multi-functionalities. We had ever applied them to other metal nanostructures and non-conjugated polymers, such as polylactic acid, dendrimers, chitosan grafted copolymer, and so on, due to their excellent bio-compatibilities. Some similar results were also obtained.

## 4. Conclusions

In summary, the well-dispersion of Pt/PVDF nanocomposite was obtained. Avoiding the aggregation of Pt nanoparticles was caused by metal nanostructure in the PVDF matrix with the in situ synthesis method. The resulting Pt/PVDF nanocomposite exhibited good photo-current switching behavior to weak NIR and visible light. The photo-current response signal depends on the content of Pt nanoparticles in the polymer. It is attributed to the optimization of interface barrier heights between metal nanoparticles and polymer matrix and reasonable charges transfer channel in metals/polymers nanocomposites. Excellent photo-switching behavior to weak NIR result from the strong interaction between Pt and PANI, promoting the separation of electrons and holes photo-induced. The resulting organic/inorganic nanocomposite also showed good tentacle sensitivity. It would be developed intelligent nanocomposite film with external stimuli responses and potential applications in flexible photodetectors in NIR, information storage, tentacle sensors, etc. This simple method is also to fabricate other kinds of organic/inorganic nanocomposites holding multi-functionalities.

## Figures and Tables

**Figure 1 materials-16-00150-f001:**
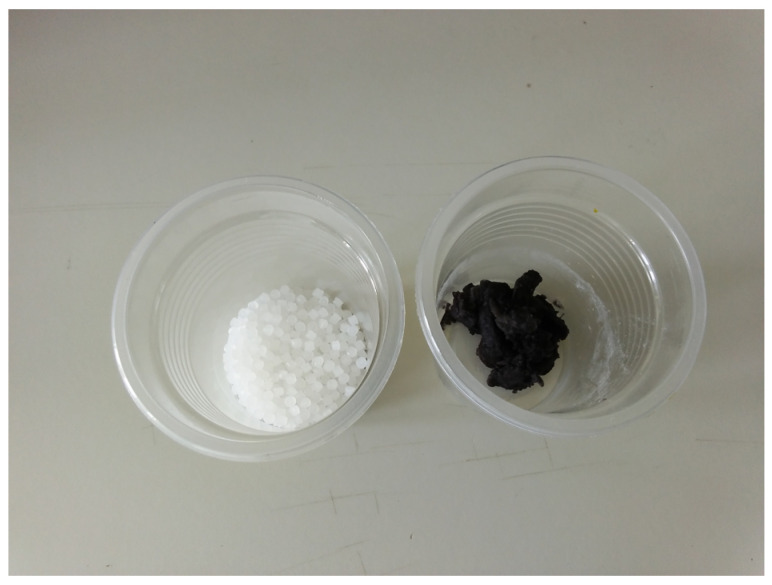
The comparative optical photos of PVDF and PVDF/Pt nanocomposite (*Left*: PVDF; *Right*: PVDF/Pt nanocomposite).

**Figure 2 materials-16-00150-f002:**
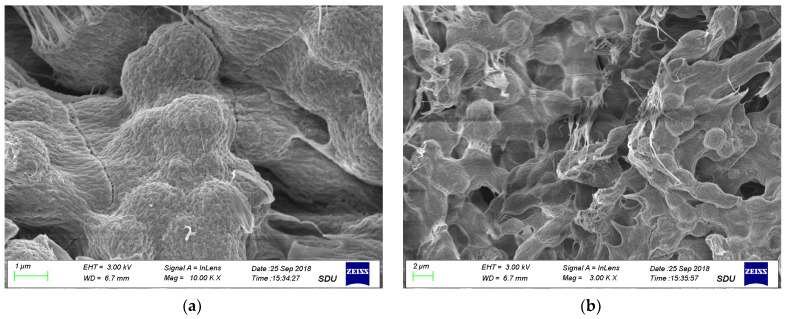
The representative SEM images of Pt/PANI/PVDF nanocomposite ((**a**): locally enlarged view of SEM; (**b**): SEM).

**Figure 3 materials-16-00150-f003:**
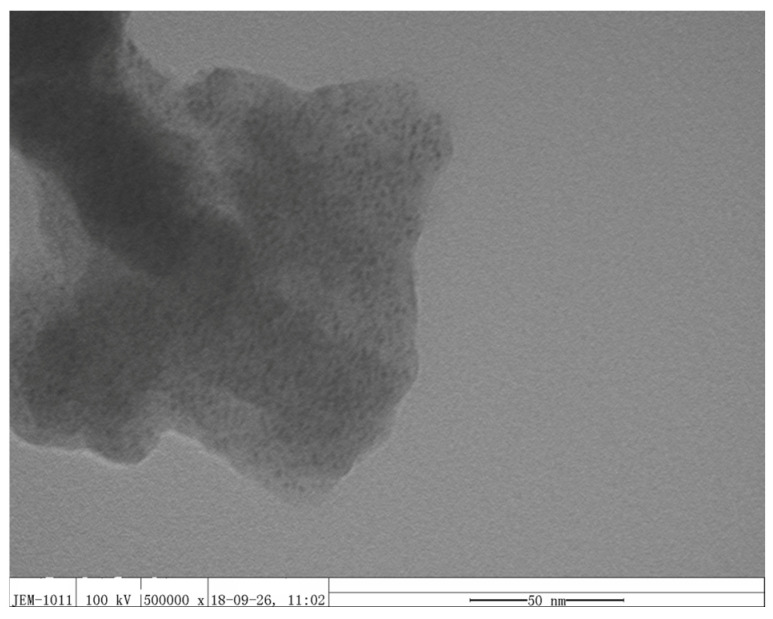
The representative TEM images of Pt/PANI/PVDF nanocomposite.

**Figure 4 materials-16-00150-f004:**
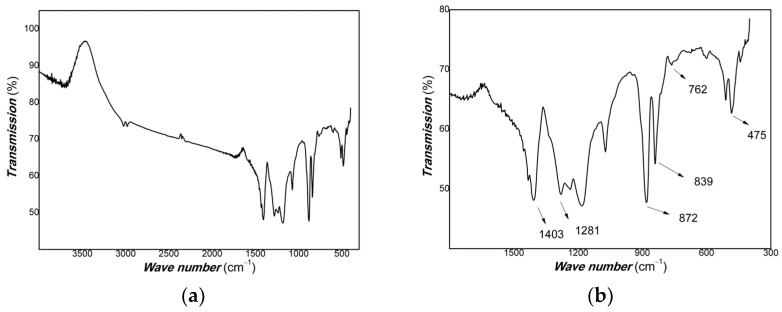
The FTIR of Pt/PVDF nanocomposite ((**a**): FTIR; (**b**): locally enlarged view of FTIR).

**Figure 5 materials-16-00150-f005:**
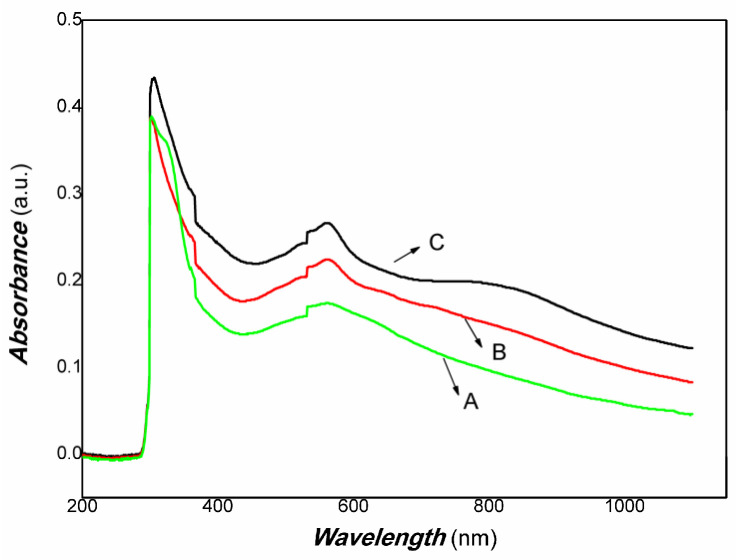
The UV-Vis-NIR of Pt/PVDF nanocomposite (A, B, C: the content of HPtCl_4_ adding is wt. 3.22, 6.25, 9.09%, respectively).

**Figure 6 materials-16-00150-f006:**
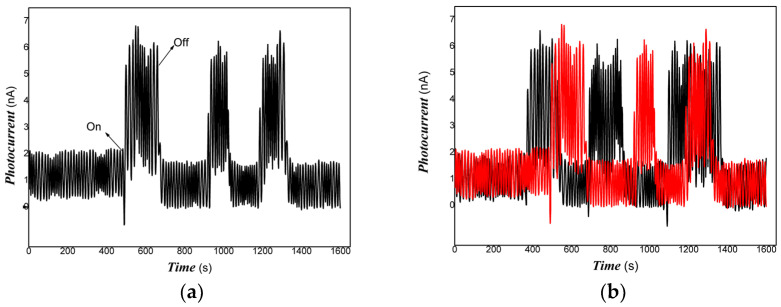
The representative photo-responses of Pt/PVDF nanocomposite to weak visible light ((**a**): one representative examination of dynamical photo-responses of Pt/PVDF nanocomposite to visible light; (**b**): two representative examinations selected of dynamical photo-responses of Pt/PVDF nanocomposite to visible light).

**Figure 7 materials-16-00150-f007:**
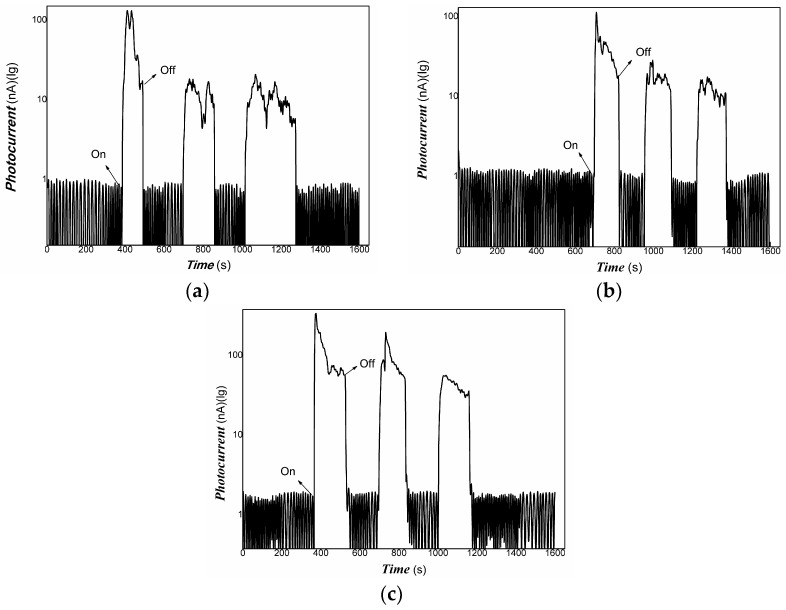
The representative photo-responses of Pt/PVDF nanocomposite to weak 808 nm NIR of 200 mW ((**a**–**c**): the content of HPtCl_4_ adding is wt. 3.22, 6.25, and 9.09% respectively).

**Figure 8 materials-16-00150-f008:**
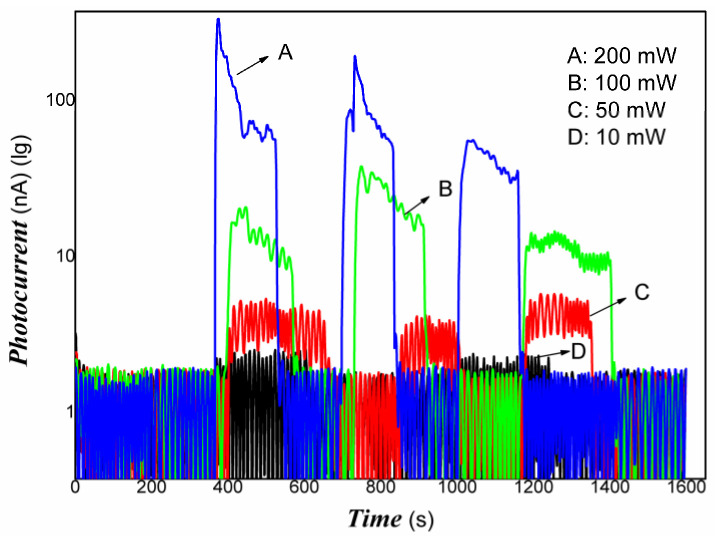
The representative photo-responses of Pt/PVDF nanocomposite to different powders of 808 nm NIR (10, 50,100, and 200 mW).

**Figure 9 materials-16-00150-f009:**
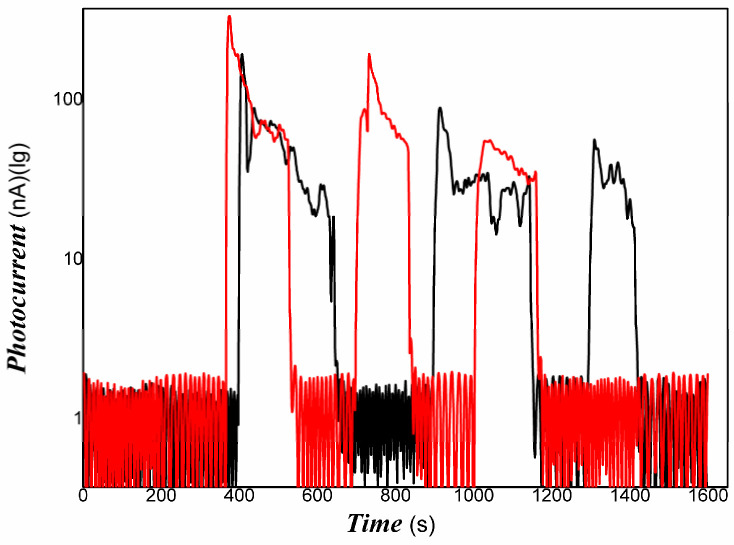
The representative repeatability of photo-responses of Pt/PVDF to weak 808 nm NIR of 200 mW (Red and black color lines respresented two representative examinations selected of dynamical photo-responses).

**Figure 10 materials-16-00150-f010:**
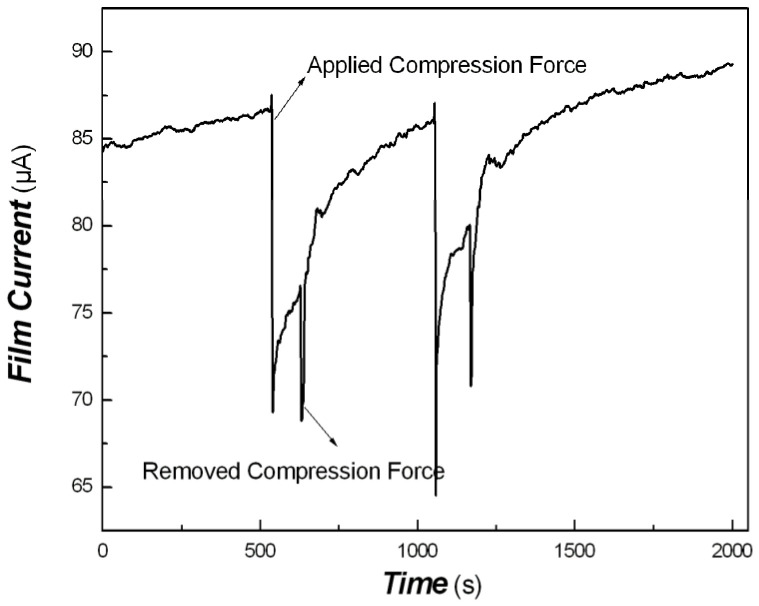
The tentacle sensitivity of Pt/PVDF nanocomposite to compression force of about 50 g.

**Figure 11 materials-16-00150-f011:**
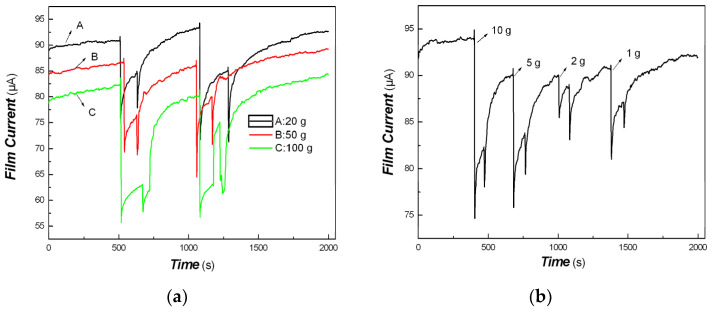
The effects of different external forces applied (1, 2, 5, 10, 20, 50, 100 g, etc.) on the tentacle sensitivity of Pt/PVDF nanocomposite ((**a**): the external forces applied is 20, 50, 100 g respectively; (**b**): the external forces applied is 1, 2, 5, 10 g respectively).

## Data Availability

The data presented in this study are available on request from the corresponding author. The data are not publicly available due to privacy.

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
