# Peer review of "Interface Optimization of Metal Quantum Dots/Polymer Nanocomposites and their Properties: Studies of Multi-Functional Organic/Inorganic Hybrid"

_materials, 2022, doi:10.3390/ma16010150_

Round 1

Reviewer 1 Report

Dear Editor

The authors have studied Interface Optimization of Metal Quantum Dots/Polymer Nanocomposites. In my opinion, the work need to be revised before its publication as below:

1-  The abstract section is too ling and broad. It should be revised to focus on the main issue, the objectives, the methodology taken, followed by the main results and conclusions.

2- Introduction is too long and diverse. The issues in the field should be clearly stated followed by the objectives and important of the proposed study.  

3- The experimental part should be combined to reduce the subheadings.

4- Conclusions need revision based on the above changes.

5- Resolution of figures has to be improved.

Author Response

Dear Sir,
Thank you for your work on our paper. The changes of paper are as follow.

Reviewer 2 Report

The work «Interface Optimization of Metal Quantum Dots/Polymer Nanocomposites and its Properties Studies of Multi-functional Organic/Inorganic Hybrid » is devoted actual and modern topic. Polymers filled with various types of fillers can be used in various fields. The authors explored the possibility of using them as flexible photodetectors in NIR, multi-functional tentacle sensors, et al. In the introduction, the authors well reveal the existing problems and the tasks to be solved in this study. A detailed literature review on the research topic is done. A detailed description of the experimental techniques allows, if necessary, to reproduce the presented results. The results are obtained using diverse and modern research methods. However, there are some recommendations:

(1) Figure 1 can hardly be called a Scheme, it does not reflect all the operations described in paragraph 2.4, therefore, it needs to be supplemented or can be deleted.

(2) For clarity, it is recommended to mark the discussed absorption bands directly on the IR spectrumÑŽ

(3) It is recommended to add a transcript to the caption of figure 5, the spectra of which are indicated by A, B, and C. Similarly, check the rest of the figures of the manuscript.

Author Response

(The authors gave the same response as above.)

Round 2

Reviewer 1 Report

The authors have performed required revisions, the paper can be accepted as it is.